# Exploring Combined Influences of Seasonal East Atlantic (EA) and North Atlantic Oscillation (NAO) on the Temperature-Precipitation Relationship in the Iberian Peninsula

Fernando S. Rodrigo 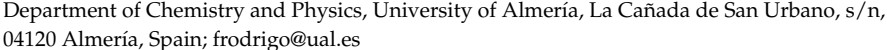

Department of Chemistry and Physics, University of Almería, La Cañada de San Urbano, s/n, 04120 Almería, Spain; frodrigo@ual.es

**Abstract:** The combined influence of the North Atlantic Oscillation (NAO) and the East Atlantic (EA) patterns on the covariability of temperatures and precipitation in 35 stations of the Iberian Peninsula during the period 1950–2019 is analysed in this work. Four EA-NAO composites were defined from teleconnection patterns' positive and negative phases: EA+NAO+, EA+NAO-, EA-NAO+ and EA-NAO-. Daily data of maximum and minimum temperature were used to obtain seasonal means (TX and TN, respectively), and the covariability of these variables with accumulated seasonal rainfall (R) was studied comparing results obtained for different NAO and EA composites. Main results indicate slight differences in the spatial coverage of correlation coefficients between R and temperature variables, except in spring when the generalised negative relationship between R and TX under EA+NAO+ and EA-NAO- disappears under EA-NAO+ and EA+NAO- composites. This result may be useful to interpret and discuss historical reconstructions of the Iberian climate.

**Keywords:** NAO; EA; temperature-precipitation covariability; Iberian Peninsula

## 1. Introduction

Atmospheric circulation patterns consist of broad and persistent patterns of atmospheric pressure anomalies (SLP) that determine the main air mass flows and large-scale advection of temperature and humidity, influencing the climate over extensive geographic regions [1–4]. Amongst these, the most important for the climate of Western Europe is the North Atlantic Oscillation (NAO), which accounts for about 40% of the variance of the SLP field in winter [5]. Its influence on air temperature, precipitation and wind speed in eastern North America and Western Europe on an inter-annual time scale has been widely recognised [6,7]. The NAO consists of a meridional dipole of sea-level pressure between the Icelandic Low and the Azores High that controls the location and intensity of western flows in the North Atlantic. The NAO directs the patterns and extremes of temperature, precipitation, snow cover and wind in Europe, especially during the winter, although its effects can spread throughout the subsequent seasons of the year [8]. Other atmospheric pressure patterns can modulate and cause temporary unseasonality in the relationships between the NAO index and climatic variables in Europe, particularly the East Atlantic pattern (EA), a low-pressure monopole to the West of Ireland located at approximately 55°N and 20–35°W [9] and representing about 16% of the variance of the SLP field [7].

The relationship between the NAO and precipitation in the Iberian Peninsula (IP) is well known [10–15]: the positive phase of the NAO is associated with stronger winds from the West, led in part by an intensification of high pressures over the IP, producing dry conditions and droughts in the area [16,17]. In contrast, the negative phase of the NAO shifts the Atlantic storms southwards, so that they invade the IP, generating heavy rains, which can sometimes cause river flooding [18–20]. Regarding temperatures, previous studies indicate that the NAO plays no role in the variability of the average IP temperature [21],

with low correlation coefficients between the NAO index and average temperatures [22], nor in temperature extremes [23]. However, some works have detected a relationship with the maximum temperatures of north-eastern Spain in winter [3] and in summer [24]. Under the NAO's positive phase, the anticyclonic conditions and clear skies contribute to an increase (decrease) in the maximum (minimum) temperatures, whereas the opposite behaviour exists under the NAO's negative phase, with cyclonic conditions and overcast skies. Merino et al. [25] found that extreme maximum temperatures are connected to the NAO's positive phase, while Mohammed et al. [26] concluded that, in years of positive (negative) NAO, the appearance of cold (warm) extremes is greater (lower).

Discussions regarding the influence of the NAO on temperatures underline the important role of the action centre locations, the Azores High and the Icelandic Low [27,28], which has been related to the NAO's interaction with other circulation patterns, in particular the EA [29–31]. Numerous studies report on the strong relationship between the EA and temperature variations in the IP throughout the year [3,30,32,33]: the EA's positive phase causes the advection of warm and humid air masses from the south and southwest over the IP, leading to an increase in temperatures. In the negative phase, the behaviour is the opposite, generating a decrease in temperatures. The EA centre is located along the NAO's nodal line, thus allowing the locations and intensities of the Icelandic Low and Azores High to modulate [34,35]. According to Comas-Bru and McDermott [29], the NAO action centres move southward (northward) when the phases of the EA and the NAO are of the same (distinct) sign. Therefore, the exclusive use of the NAO does not seem sufficient to explain the climatic variability in the area, since its impacts are modulated by the sign of the EA [8]. In fact, the interaction between the NAO and the EA may have played an important role in the transition between different climatic phases, such as the Medieval Warm Period and the Little Ice Age [31,34].

In terms of impacts, the covariability of temperatures and rains may be more important than changes in one or other variable individually [36–40]. Changes in precipitation come from local correlations with temperature changes, mainly due to the thermodynamic relationships between the two variables [41,42]. Temperature is one of the best studied climatic variables, whereas our ability to predict precipitation is limited due to complex interactions between multiple factors. Therefore, characterising the covariability between precipitation and temperature can improve our understanding of precipitation behaviour, as well as the joint impact of both variables [40]. The covariability between surface temperature and precipitation, and how it responds to climate variability, is not well understood [43–47]. Large-scale circulation can alter (by amplifying or reducing) the relationship between temperature and precipitation [38,48]. Consequently, changes in teleconnection patterns can alter this relationship in the IP, particularly in winter, when the NAO's influence is greater.

As far as we know, only a few papers have studied the possible influence of atmospheric circulation patterns on temperature and precipitation covariability: Beniston and Goyette [49] studied the local influence of NAO on combined temperature and precipitation modes in Switzerland, and López-Moreno et al. [2] used a similar methodology to study combined modes of temperature and precipitation in mountainous areas of the Mediterranean Basin. Luoto and Nevalainen [50], using proxy data from northern Europe, studied changes in the temperature-precipitation ratio over the past two millennia and their relationship to the NAO phases, identifying warm-dry (cold-humid) periods in areas where the positive (negative) phase of the NAO predominated; Zubiate et al. [7] studied the combined influence of NAO and EA patterns on climate variability in Western Europe, and Sánchez-López et al. [31] analysed the role of the NAO, EA and their interactions, on the climate variability observed in the IP over the past 2000 years.

The IP climate is remarkably sensitive to changes in atmospheric circulation configuration due to its complex orography and its location in a transition region between the mid and subtropical latitudes, and between the Atlantic Ocean and the Mediterranean Sea [51]. These conditions give rise to many microclimatic regimes, which have varied sensitivity to the main atmospheric circulation patterns [33]. Changes in the relationship between

temperature and precipitation can be directed both by circulation patterns and by local weather factors [47]. Therefore, it seems advisable to analyse the influence of the combined modes of EA and NAO circulation on temperature and precipitation covariability at the local level, and a posteriori to analyse the special coverage of such influences. Only a few studies have considered the local influence of circulation patterns on combined temperature and precipitation modes [2]. Our work uses the series of 35 weather stations evenly distributed across the IP for this purpose. Daily analysis is complicated due to the data noise that appears at this time resolution, weakening the correlations between temperature and precipitation [40]. Therefore, the objective of this work is to analyse the influence of the combined modes of the distinct EA and NAO phases on temperature and precipitation covariability over a seasonal time scale; as far as we are aware, this is a goal that, to date, has hardly been studied in the specialised literature.

## 2. Data and Methods

Monthly EA and NAO indices were obtained from the Climate Prediction Center [52]. Monthly values for the period 1950–2019 were averaged to obtain a seasonal index, defining the seasons of the year in the usual way: Winter (December-January-February), Spring (March-April-May), Summer (June-July-August) and Autumn (September-October-November). For each season of the year and for each index the median of the entire series was estimated, $EA_{med}$ and $NAO_{med}$. Table 1 shows the values obtained.

**Table 1.** Median of the circulation indices EA and NAO for each season of the year during the period 1950–2019.

| Mode | Winter | Spring | Summer | Autumn |
|:---:|:---:|:---:|:---:|:---:|
| EA | −0.37 | −0.13 | −0.06 | −0.16 |
| NAO | −0.23 | −0.14 | +0.02 | +0.17 |

The median was used to determine the distinct phases of the circulation pattern, EA+ = {EA/EA > $EA_{med}$}, EA- = {EA/EA < $EA_{med}$ NAO+ = } NAO/NAO > $NAO_{med}$} and NAO- = {NAO/NAO < $NAO_{med}$}. Although this criterion includes 'normal' and extreme values in each phase under the same category, it allows one to obtain subsets with sufficient data to attempt statistical approximations. Figure 1 shows the winter series of both patterns as an example.

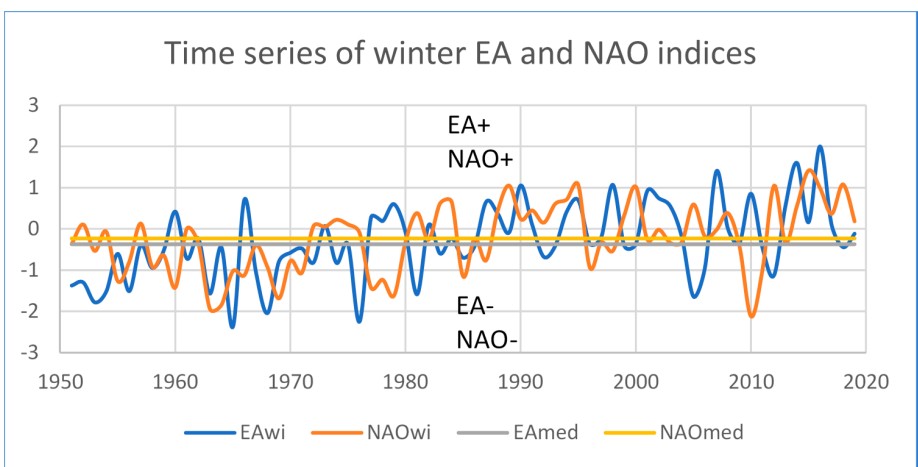

**Figure 1.** EA and NAO winter indices, period 1950–2019. Median values ($EA_{med}$ and $NAO_{med}$) are indicated.

Subsequently, 4 data subsets were determined, paying attention to the simultaneous occurrence of the different phases, i.e., EA + NAO +, EA+NAO-, EA-NAO+ and EA-NAO-

for each season of the year. Table 2 shows the distribution of years in each of the subsets for each season of the year. The number of years in each subset is of the order of 17 in winter and spring, with a uniform distribution over the four subsets. In summer and autumn there is a slight predominance of the EA+NAO- and EA-NAO+ phases, at 21 years, compared to 14 for the other two phases. The series obtained were tested to check if the amount of data was sufficient, obtaining satisfactory results. Bastos et al. [8] conducted a similar study for the 1982–2012 period using the indices corresponding to the intra-annual period between December and April. They defined the positive (negative) phases of each mode considering the years when the index was greater (less) than the upper (lower) tercile. The result is that the amount of data for each phase is evidently low (for example, only 4 years were categorised as the EA+NAO+ phase). Even though this definition allows one to focus on the years with extreme phases, it introduces a new category, which these authors call "neutral", corresponding to those index values between the lower and the upper tercile. Despite the differences in the seasonal definition and the four combined EA-NAO modes, over the 31 years that are common to the Bastos et al. study and to this work, coincidences are found for 18 years (58% of cases).

**Table 2.** Years corresponding to the four EA-NAO composites.

| Season | EA+NAO+ | EA+NAO- | EA-NAO+ | EA-NAO- |
|---|---|---|---|---|
| Winter | 1973,1984,1988,1989 1990,1991,1994,1995 2000,2002,2007,2008 2014,2015,2016,2017 2019 | 1960,1962,1966,1977 1978,1979,1980,1982 1987,1996,1997,1998 2001,2003,2004,2009 2010,2013 | 1952,1954,1957,1961 1972,1974,1975,1976 1981,1983,1992,1993 1999,2005,2006,2012 2018 | 1951,1953,1955,1956 1958,1959,1963,1964 1965,1967,1968,1969 1970,1971,1985,2011 |
| Spring | 1959,1969,1986,1989 1992,1994,2000,2002 2003,2004,2007,2009 2014,2015,2016,2017 2018 | 1950,1952,1961,1964 1970,1973,1977,1979 1941,1983,1988.1998 2001,2005,2006,2008 2010,2013 | 1954,1956,1960,1963 1967,1972,1974,1976 1978,1982,1985,1987 1990,1991,1993,1997 2011,2012 | 1951,1953,1955,1957 1958,1962,1965,1966 1968,1971,1975,1980 1984,1995,1996,1999 2019 |
| Summer | 1961,1981,1988,1990 1992,1994,1999,2002 2003,2005,2013,2017 2018 | 1950,1951,1952,1958 1982,1985,1998,2000 2001,2004,2006,2007 2008,2009,2010,2011 2012,2014,2015,2016 2019 | 1953,1955,1959,1964 1965,1967,1970,1971 1972,1973,1975,1976 1978,1979,1983,1984 1989,1991,1995,1996 1997 | 1954,1956,1957,1960 1962,1963,1966,1968 1969,1974,1977,1980 1987,1993 |
| Autumn | 1951,1954,1969,1979 1982,1984,1986,1999 2009,2011,2014,2015 2016,2018 | 1960,1968,1970,1973 1980,1981,1983,1985 1987,1996,1998,2000 2001,2003,2005,2006 2010,2012,2013,2017 2019 | 1953,1956,1957,1958 1959,1961,1963,1964 1967,1971,1972,1974 1975,1977,1978,1989 1990,1991,1993,2007 2008 | 1950,1952,1955,1962 1965,1966,1976,1988 1992,1994,1995,1997 2002,2004 |

The database used in this study comprises the daily precipitation and the maximum and minimum daily temperatures of 35 locations, covering the main IP climate domains during the 1950–2019 period. The data series were obtained from the European Climate Assessment & Dataset Project (ECA&D, [53,54]). The data and metadata are available at http://www.ecad.eu (accessed on 28 October 2020). This database contains 220 Spanish and 39 Portuguese stations. The time series of a station and a particular season was rejected if it had more than 9 days (about 10%) with data gaps. For each station, time series of a season of the year was removed if it had more than six data gaps (about 10%). These restrictions were applied simultaneously to temperature and precipitation data. As a result, the number of selected stations was reduced to 35. Unfortunately, only two Portuguese stations (Lisbon and Braganza), survived after the application of these criteria. This database was described in a previous work on coherent variability between temperatures and seasonal precipitation in the IP [55]. Figure 2 shows the location of the weather stations and Table 3 shows their

main geographic data, along with the percentage of days without data. Even though the set of stations is small, it is representative of the range of IP climate regimes [56], including stations on the north coast, Mediterranean coast and the central, western and southern areas of the IP. The total seasonal precipitation (sum of daily precipitation for each season, R), average minimum daily temperatures (TN) and average maximum daily temperatures (TX) were obtained from the daily data.

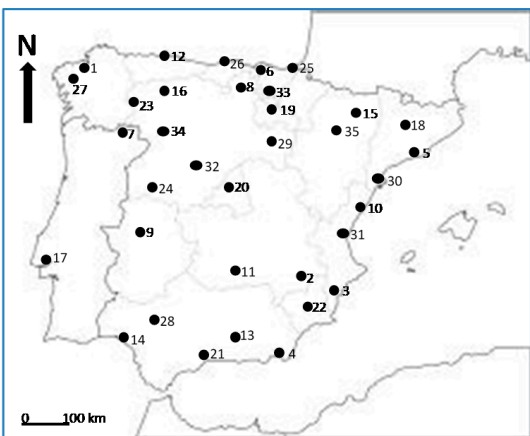

**Figure 2.** Map with the 35 stations selected In the Iberian Peninsula (numerical code in Table 1 is included).

**Table 3.** Meteorological stations studied in this work (data from 1950–2019).

| Code | Station | Latitude | Longitude | Height (m asl) | Gaps (days, %) |
|------|---------|----------|-----------|----------------|----------------|
| 1 | A Coruña | 43°22′N | 08°23′W | 21 | 0.00 |
| 2 | Albacete | 38°59′N | 01°51′W | 681 | 0.20 |
| 3 | Alicante | 38°20′N | 00°28′W | 5 | 0.04 |
| 4 | Almería | 36°50′N | 2°27′W | 16 | 0.10 |
| 5 | Barcelona | 41°22′N | 02°10′E | 13 | 0.30 |
| 6 | Bilbao | 43°15′N | 02°57′W | 6 | 0.90 |
| 7 | Braganza | 41°48′N | 06°45′W | 700 | 1.40 |
| 8 | Burgos | 42°20′N | 03°41′W | 859 | 0.07 |
| 9 | Cáceres | 39°28′N | 06°22′W | 457 | 0.00 |
| 10 | Castellón | 39°58′N | 00°03′W | 27 | 0.70 |
| 11 | Ciudad Real | 38°59′N | 03°55′W | 625 | 0.05 |
| 12 | Gijón | 43°32′N | 05°42′W | 3 | 0.07 |
| 13 | Granada | 37°10′N | 03°36′W | 684 | 0.70 |
| 14 | Huelva | 37°15′N | 06°57′W | 24 | 0.30 |
| 15 | Huesca | 42°08′N | 00°24′W | 483 | 2.00 |
| 16 | León | 42°35′N | 05°34′W | 837 | 0.20 |
| 17 | Lisboa | 38°43′N | 09°10′W | 2 | 5.00 |
| 18 | Lleida | 41°37′N | 00°38′E | 167 | 0.01 |
| 19 | Logroño | 42°28′N | 02°26′W | 384 | 0.02 |

**Table 3.** *Cont.*

| Code | Station | Latitude | Longitude | Height (m asl) | Gaps (days, %) |
|------|---------|----------|-----------|----------------|----------------|
| 20 | Madrid | 40°25′N | 03°41′W | 657 | 0.00 |
| 21 | Málaga | 36°43′N | 04°25′W | 8 | 0.40 |
| 22 | Murcia | 37°59′N | 01°07′W | 42 | 0.80 |
| 23 | Ponferrada | 42°32′N | 06°35′W | 512 | 0.10 |
| 24 | Salamanca | 40°57′N | 05°39′W | 798 | 0.05 |
| 25 | San Sebastián | 43°19′N | 01°59′W | 7 | 0.05 |
| 26 | Santander | 43°28′N | 03°48′W | 8 | 0.05 |
| 27 | Santiago | 42°53′N | 08°32′W | 260 | 0.20 |
| 28 | Sevilla | 37°23′N | 05°59′W | 11 | 0.30 |
| 29 | Soria | 41°46′N | 02°28′W | 1061 | 0.30 |
| 30 | Tortosa | 40°48′N | 00°31′E | 14 | 0.00 |
| 31 | Valencia | 39°28′N | 00°22′W | 16 | 0.30 |
| 32 | Valladolid | 41°39′N | 04°43′W | 690 | 0.20 |
| 33 | Vitoria | 42°50′N | 02°40′W | 539 | 1.10 |
| 34 | Zamora | 41°29′N | 05°45′W | 649 | 1.30 |
| 35 | Zaragoza | 41°39′N | 00°53′W | 208 | 0.02 |

The database was divided into four subsets, corresponding to the combined phases of the EA and NAO patterns. The objective was to compare the distributions resulting from this division. For each subset we can consider the temperature and precipitation data as a bivariate distribution, characterised by its vector of average values and its covariance matrix [57]. As a first approximation, the behaviour of the mean values and the correlation coefficients between the two variables were studied for each of the subsets. The mean temperature and precipitation values for each phase were compared to the mean of the entire 1950–2019 period using the t-test for the difference between the means, at a confidence level of 95%. For each seasonal variable X (X = R, TN, TX) was calculated the difference $\overline{X(\text{EAiNAOj})} - \overline{X}$, where $\overline{X}$ is the seasonal mean for the entire period, and the subindices i and j express the different phases (+, − of the circulation indices. A standard 30-year reference period was not used because the temporal evolution of the EA and NAO indices (predominantly one phase or the other depending on the chosen period) could have skewed the results. The results were mapped to explore the spatial coverage of significant differences. The correlation coefficients between TN, TX and R were calculated for each of the phases, EA+NAO+, EA+NAO-, EA-NAO+ and EA-NAO-, using the 95% confidence level, and the results were mapped to look for spatial differences and similarities.

## 3. Results

Firstly, we analysed the behaviour of the average R, TN and TX values for each season of the year and each EA-NAO combined phase by comparing these average values with seasonal values for the entire period (1950–2019). Next, the R-TN and R-TX correlations were studied for each season of the year and each EA-NAO combined phase. The results from this study are summarised in 48 maps for the analysis of the average values (3 variables, 4 modes and 4 seasons of the year), and 32 maps for the study of the correlations (2 correlations, 4 modes and 4 seasons of the year). For obvious reasons of limited space, only the most significant results are displayed.

### 3.1. Average Values

Figure 3 shows the behaviour of the average winter precipitation values for each of the 4 EA-NAO combined modes. Unsurprisingly, a decrease (increase) in precipitation is observed under the positive (negative) phase of NAO. The map corresponding to the EA+NAO− and EA-NAO+ phases shows the known result of the NAO's influence on the IP, with significant results in the central-western sector and a slight (but not significant) influence on the Mediterranean coast and the north coast.

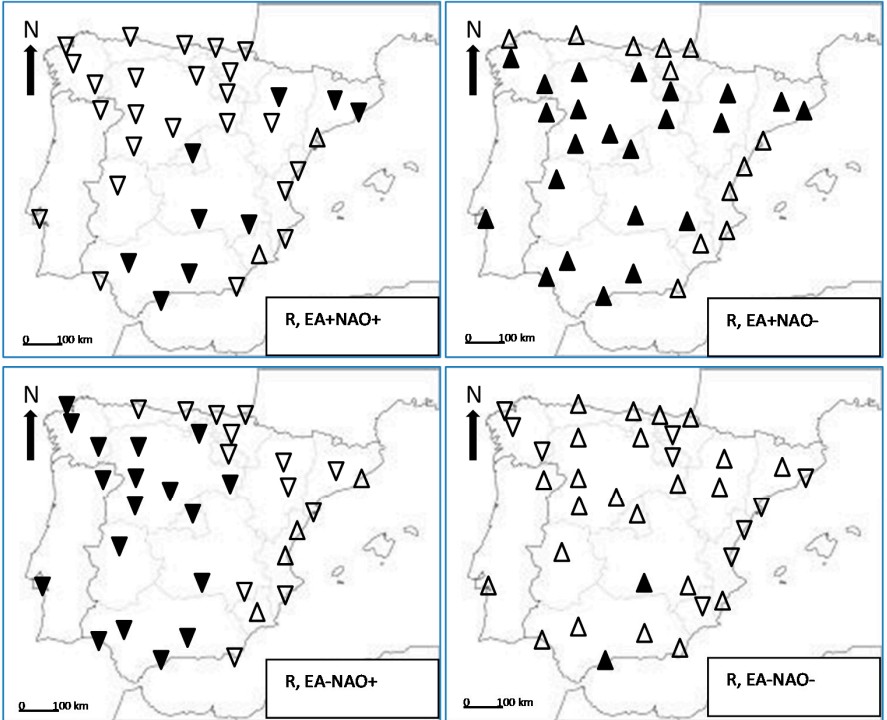

**Figure 3.** Difference of R mean values for each EA-NAO composite and the mean of the complete period 1950–2019 for winter. Triangle up (down) means positive (negative) difference, full (empty) triangle means significant (not significant) difference at the 95% confidence level.

The EA+ phase (EA-) indicates the existence of a low (high) pressure centre in the North Atlantic, with advection (blocking) of south-southwesterly humid air over the IP, which reinforces the role of the opposite phases of the NAO. A similar result was found by Bastos et al. [8] in their behavioural analysis of the EA and NAO modes in antiphase, and by Manzano et al. [17] in their study on the impact of circulation patterns on the SPEI index for the IP. In the case of the EA-NAO- mode, the differences stop being significant even in the central-western sector, except in two locations. The EA- phase involves the existence of a blocking high in the Atlantic, which would inhibit the increase in precipitation under NAO-. On the other hand, the EA phase appears to influence the spatial coverage of decreased precipitation under NAO+, with a shift from East (EA+NAO+) to West (EA-NAO+). The NAO's influence on IP precipitation is reinforced when it is in the opposite phase to that of the EA and is weakened if both modes are in the same phases.

Figures 4 and 5 show the analytical results for the winter minimum (TN) and maximum (TX) temperatures, respectively. The EA+ pattern is associated with an increase in temperatures across Europe [4] and particularly in the IP [3,30,31]. Under EA+, there is advection of warm and humid air over the IP, resulting in increased temperatures, while the opposite behaviour is to be expected under EA-. One can observe that, while the differences are significant for the opposing phases of TN, the opposite occurs for TX. TN exhibits similar behaviour to R as a result of rising minimum temperatures under overcast skies.

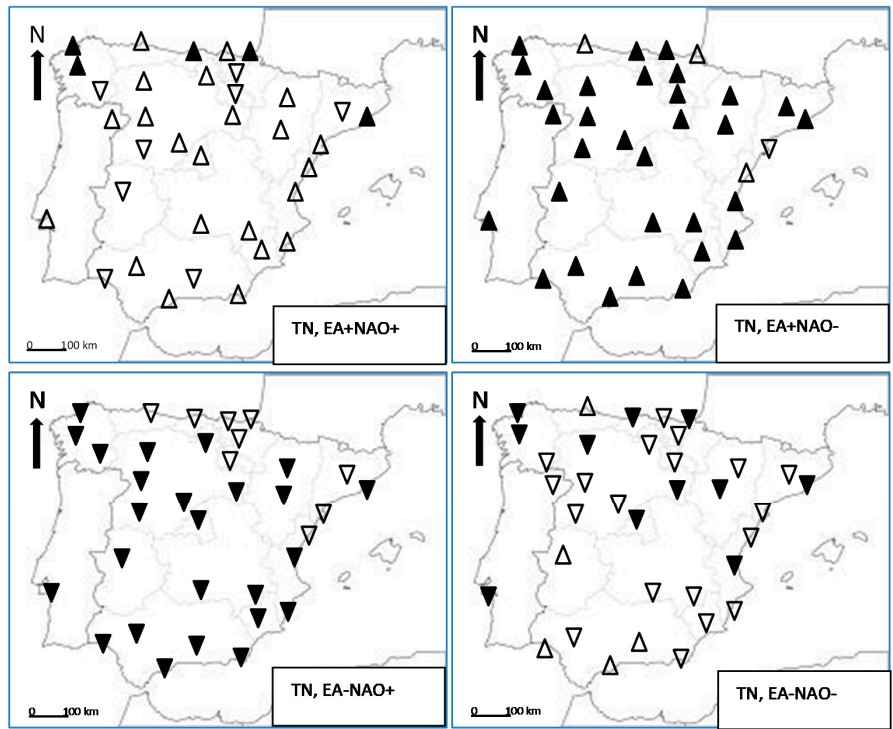

**Figure 4.** Difference of TN mean values for each EA-NAO composite and the mean of the complete period 1950–2019 for winter. Triangle up (down) means positive (negative) difference, full (empty) triangle means significant (not significant) difference at the 95% confidence level.

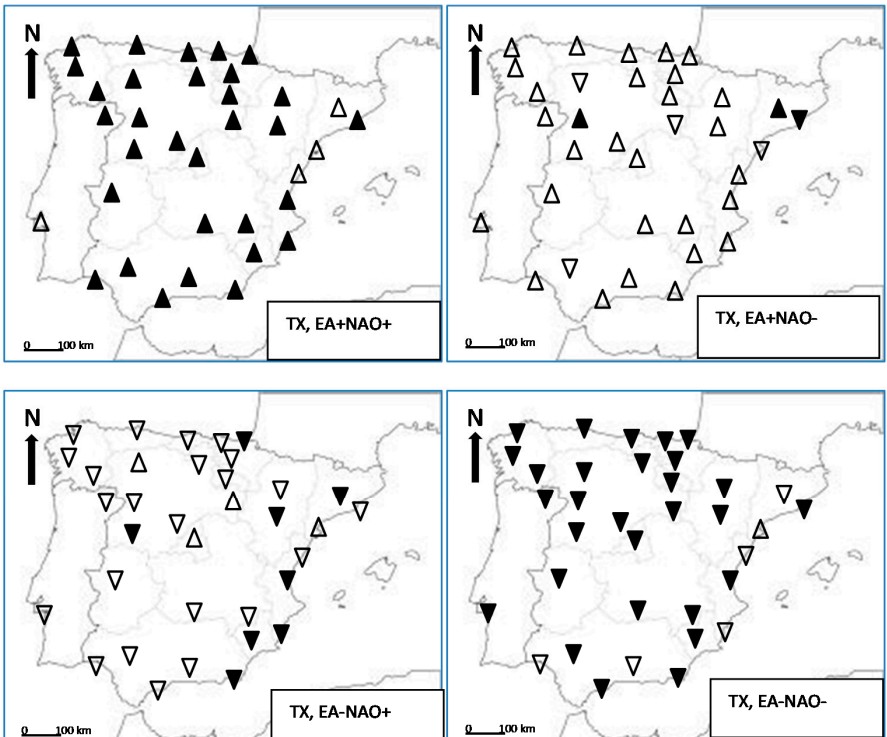

**Figure 5.** Difference of TX mean values for each EA-NAO composite and the mean of the complete period 1950–2019 for winter. Triangle up (down) means positive (negative) difference, full (empty) triangle means significant (not significant) difference at the 95% confidence level.

In the case of maximum temperatures, the increase (decrease) of temperatures under EA+ (EA-) is reinforced by anticyclonic (cyclonic) conditions under NAO+ (NAO-) [25]. Conversely, when the phases of the EA and NAO are opposite, the differences are not significant (except in a few locations). This behaviour, of intensified (reduced) anomalies when the two modes have the same (different) sign, has already been explained by Comas-Bru and McDermott [29]. Zubiate et al. [7] found similar behaviour in their study looking at the influence of the EA and NAO modes on wind speed in the IP, as did Bastos et al. [8] in their behavioural analysis of the PDSI index for each of the 4 combined EA-NAO phases.

In the winter months, the atmosphere is more dynamically active and, as a result, the NAO's influence on surface temperature and precipitation is greater whereas it is much less during the other seasons [6]. As a result, no significant differences in R were found in spring, summer and autumn under any of the 4 EA-NAO combined modes.

If the EA modulates the influence of the NAO on winter precipitation, for the other seasons, it appears that the NAO modulates the influence of the EA on temperatures. Figure 6 shows the TN differences in spring for the EA+NAO+ and EA-NAO- cases. Here, we see that the differences are only significant at a few stations in the eastern IP sector. Even though the EA has a special broad pattern across the IP with an increase (decrease) under EA+ (EA-) [3], this result shows how for minimum temperatures, as in winter, the NAO's role is to reduce these differences, since the TN decreases (increases) under NAO+ and clear skies (NAO- and overcast skies).

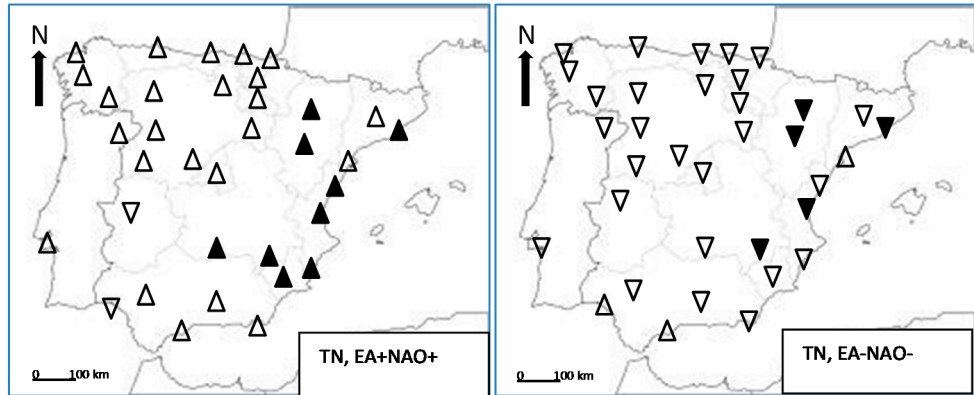

**Figure 6.** Difference of TN mean values for EA+NAO+ and EA-NAO- composites and the mean of the complete period 1950–2019 for spring. Triangle up (down) means positive (negative) difference, full (empty) triangle means significant (not significant) difference at the 95% confidence level.

Figure 7 shows the behaviour of TN and TX in autumn in EA-NAO+ mode. The negative EA phase induces a decrease in TN and TX, which is slightly reinforced (reduced) in TN (TX) by the influence of NAO+. Indeed, in the case of TX under EA-NAO+, this compensatory effect is detected in the western IP sector, while on the Mediterranean coast and the Cantabrian coast in the north, where the NAO influence is less, the significant differences caused by the EA are maintained. According to Kenawy et al. [3] the relationship between the EA pattern and autumn temperatures is closer in the vicinity of the Mediterranean and Cantabria than in the IP interior. This same behaviour (increased TN and TX under EA+, decreased under EA-), with a variation in the spatial distribution of significant differences under the different NAO phases, was also found in summer (not shown).

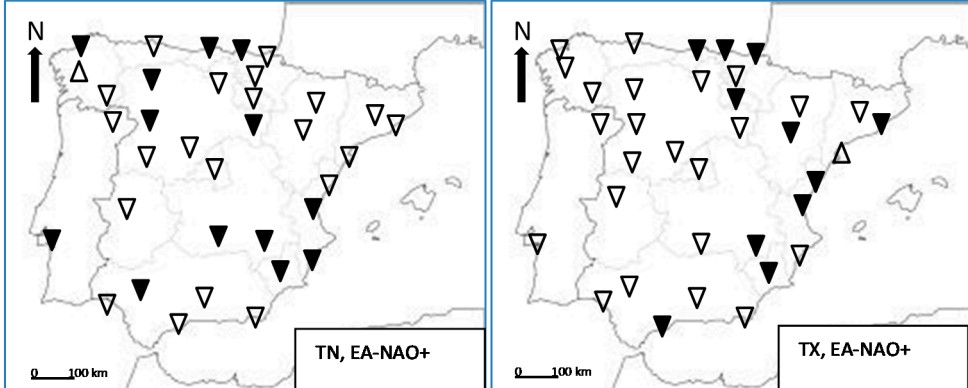

**Figure 7.** Difference of TN and TX mean values for EA-NAO+ composite and the mean of the complete period 1950–2019 for autumn. Triangle up (down) means positive (negative) difference, full (empty) triangle means significant (not significant) difference at the 95% confidence level.

*3.2. Correlations*

In the following section, and as a first approximation to this study, we will focus on the possible changes in correlation coefficients, from significant to non-significant values (at a 95% confidence level) or vice versa, without regard to the magnitude of these changes, a theme we will take up in future work.

As we have seen, there is some interconnection between the EA and NAO that influences precipitation behaviour and winter temperatures. We wonder if the same is true of the correlations. Figure 8 shows the correlation coefficients between winter temperatures and accumulated precipitation under the EA-NAO+ and EA+NAO+ subsets. First, one observes that the R-TN (R-TX) correlations are positive (negative) as a result of the relationship between temperatures, radiative balance and cloud cover [55,58]. However, the number of stations where these correlations are significant is small, especially in the case of TX (only 7 stations (20%) with significant coefficients under the EA-NAO+ and none under EA+NAO+). No spatial pattern is seen changing from one subset to another, if anything there is a weakening of the negative R-TX correlations in the case of EA+NAO+. Except for differences in some specific locations, the conclusion is that one cannot infer that switching from one subset to other results in important modifications in the correlation between temperatures and precipitation. The same results were obtained for the other combinations of winter EA and NAO modes (not shown).

Figure 9 shows the results for the spring analysis. The positive relationship found between TN and R in winter weakens in spring and is even reversed to a negative correlation in some seasons. Soil-atmosphere moisture feedback has been cited as the main candidate to explain this behaviour: lower (higher) precipitation is associated with reduced (increased) soil humidity and latent heat flow, and therefore an increase (decrease) in surface warming, resulting in an increase (decrease) in temperatures [38,55,59,60]. According to our results, this mechanism is intensified when the EA and NAO are in the same phase, although significant R and TN correlations appear at only 6 stations (17%) under EA+NAO+, and at 10 stations (29%) under EA-NAO-. In the case of maximum TX temperatures, this result is generalised to 23 stations (66%) in the IP under EA+NAO+ and to 31 stations (89%) under EA-NAO-. When the two indices are of the same sign, there is an increase in the magnitude of the anomalies associated with SLP [8,34], and, as shown in Figure 9, an intensification of the negative correlation between TX and R. This negative correlation leads to the predominance of warm-dry (EA+NAO+) or, alternatively, cold-humid (EA-NAO-) conditions. Zubiate et al. [7]) also found an intensification (weakening) of the wind speed in the IP when the sign of the two patterns was the same (opposite).

Sánchez-López et al. [31] report the predominance of the EA+NAO+ phase during the Medieval Climatic Anomaly (MCA, 900–1300 AD) and of the EA-NAO- phase during the Little Ice Age (LIA, 1300–1850). In fact, the LIA has commonly been associated with cold

and humid conditions in the IP [61,62]. These results can be compared with the analysis of proxy data (tree rings, documentary data) for this climatic period, which mostly report on spring weather conditions [63,64]. Documentary data are obtained from historical records, mainly concerned with agricultural production, especially wheat, which is harvested in late spring and early summer. For this reason, the weather conditions in spring received special attention. In an earlier work [64] a predominance of cold and humid springs was detected in the south of the IP during the 1792–1808 period, in the middle of the LIA, indicating a negative relationship between rainfall and temperatures. Therefore, we can interpret that past conditions (cold and wet springs) were the result of the predominance of the EA-NAO- phase.

Figure 10 shows the R-TX correlations corresponding to summer. The results are very similar to those for spring, with a higher percentage of significant correlations when the two modes are in the same phase. Under the EA-NAO- phase, this influence appears to be restricted to the northern half of the IP, which may be the result of precipitation scarcity in the south and on the Mediterranean coast of the IP during this season of the year. Some positive but not fundamentally significant correlations appear at stations in the southern sector and on the Mediterranean coast. One possible explanation is the growing influence of convective rain (summer storms) at this time of year [55]. The autumn results (not shown) are inconclusive, as the number of seasons with significant correlations under the four combined modes is noticeably low. Positive correlations between R and TN already appear, indicating the transition to winter conditions.

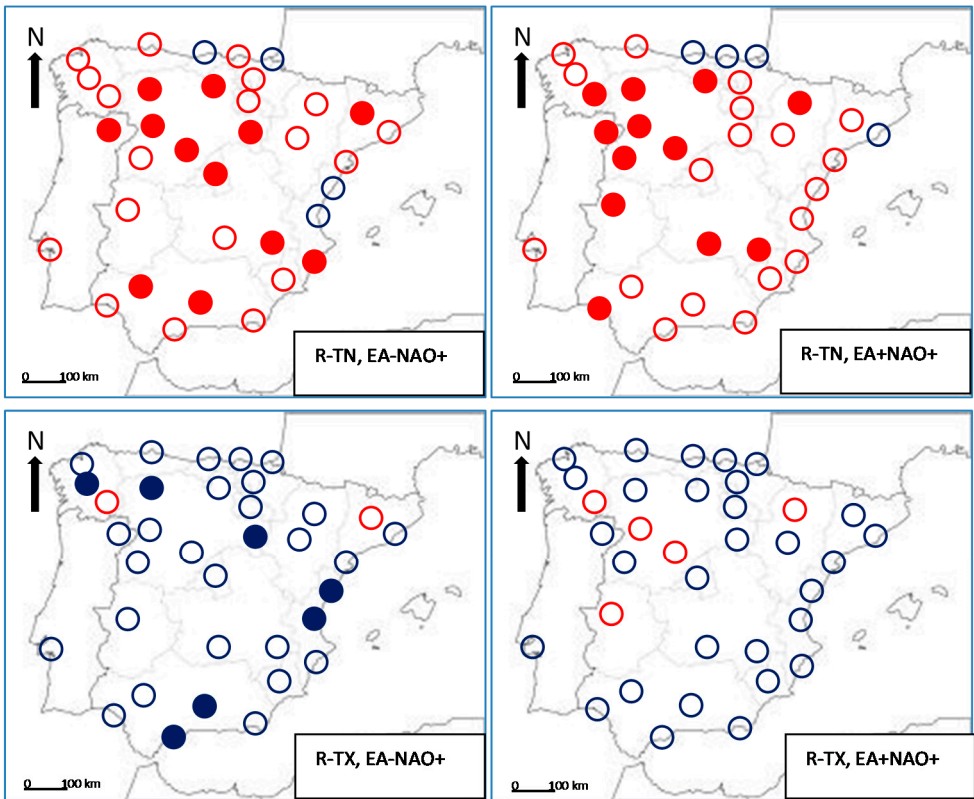

**Figure 8.** Correlation coefficients between TN, TX and R under EA-NAO+ and EA+NAO+ composites for winter. Red: positive; Blue: negative; Full (empty): significant (non-significant) at the 95% confidence level.

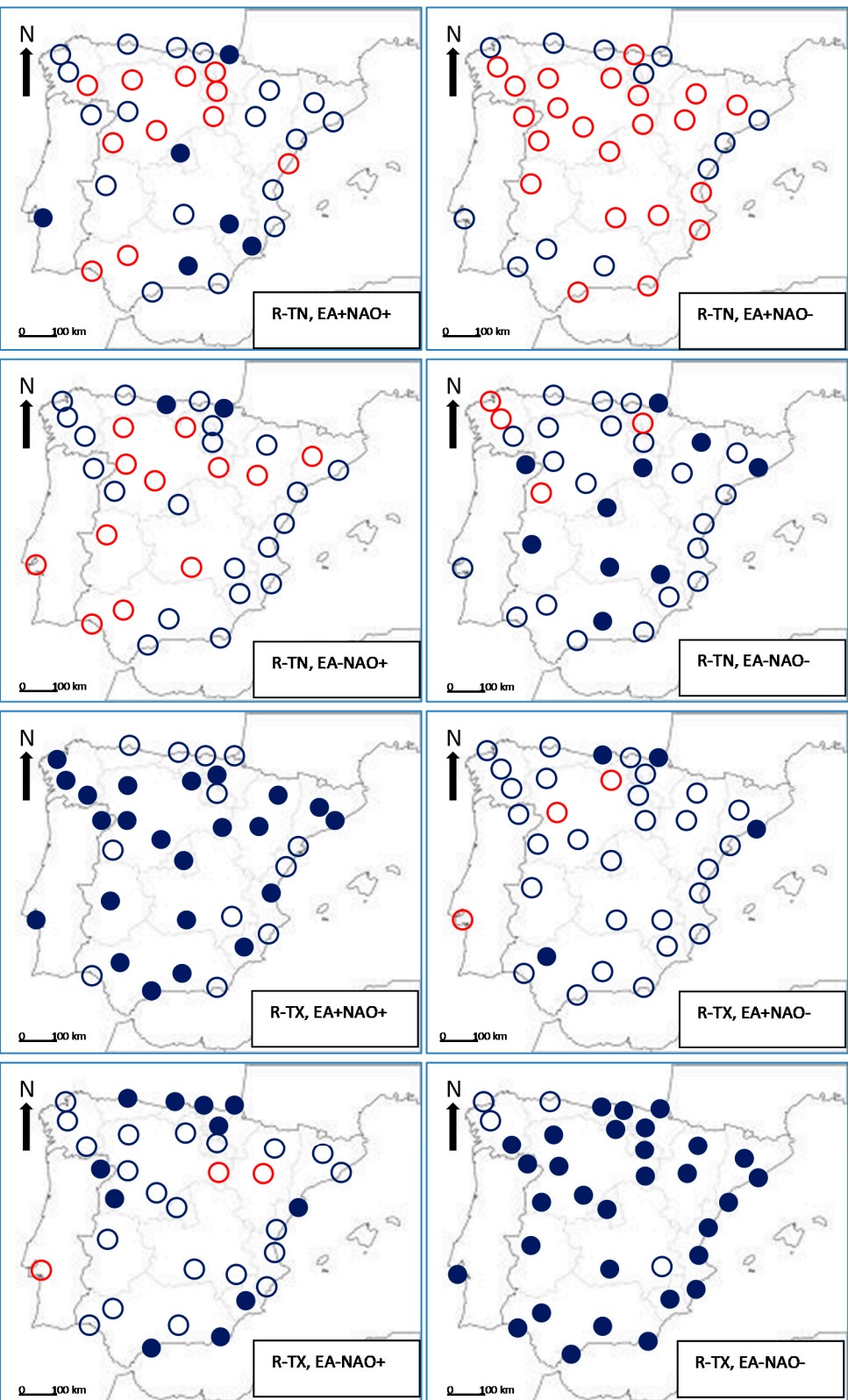

**Figure 9.** Correlation coefficients between TN, TX and R under the four composites EA+NAO+, EA+NAO-, EA-NAO+ and EA+NAO+ for spring. Red: positive; blue: negative; full (empty): significant (non-significant) at the 95% confidence level.

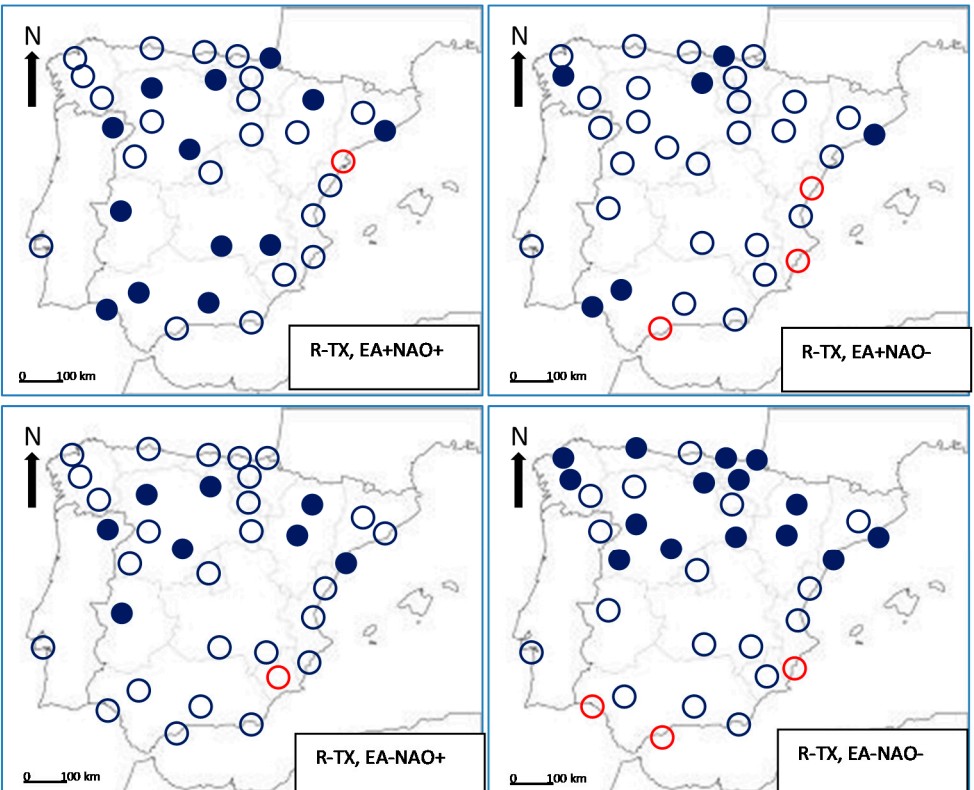

**Figure 10.** Correlation coefficients between TX and R under the four composites EA+NAO+, EA+NAO-, EA-NAO+ and EA+NAO+ for summer. Red: positive; blue: negative; full (empty): significant (non-significant) at the 95% confidence level.

## 4. Conclusions

The main results of this study can be summarised as follows:

- In winter, the influence of the NAO on precipitation and minimum temperatures across the IP is reinforced when it is in the opposite phase to that of the EA and weakens if both modes are in the same phases. With respect to maximum temperatures, anomalies intensify (reduce) when the two modes have the same (different) sign.
- No significant differences in precipitation were found in spring, summer or autumn under any of the 4 combined EA-NAO modes.
- If the EA modulates the influence of the NAO in winter precipitation, for the other seasons, it appears that the NAO modulates the influence of the EA on temperatures. The increase (decrease) of TN and TX under EA+ (EA-), with variation in the spatial distribution of significant differences under the different NAO phases, was found primarily in spring and summer.
- It cannot be inferred from the comparison between different subsets the appearance of major changes in the correlation between autumn and winter temperatures and precipitation.
- In spring and summer, when the two indices have the same sign, the negative correlation between TX and R intensifies. This negative correlation leads to the predominance of warm-dry (EA+NAO+) or, alternatively, cold-humid (EA-NAO-) conditions. This result can help in the interpretation of climatic reconstructions based on IP proxy data.

A complete explanation of climate variability in the IP requires the role of other circulation patterns to be taken into account, in particular the Scandinavian pattern (SCAND, [29,32,33] and the Western Mediterranean Oscillation (WeMO, [65]). Analysing the role of these patterns on the covariability of temperature and precipitation in the IP will be the subject of future works.

**Funding:** This research received no external funding.

**Institutional Review Board Statement:** Not applicable.

**Informed Consent Statement:** Not applicable.

**Data Availability Statement:** Monthly EA and NAO indices were obtained from the Climate Prediction Center (available at https://www.ncep.noaa.gov/teledoc, accessed on 28 October 2020). IP data series were obtained from the European Climate Assessment & Dataset Project (ECA&D, data and metadata are available at http://www.ecad.eu, accessed on 28 October 2020).

**Acknowledgments:** The author wishes to express his gratitude to the anonymous referees for their useful comments.

**Conflicts of Interest:** The author declares no conflict of interest.

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
