# Peer review of "Exploring Combined Influences of Seasonal East Atlantic (EA) and North Atlantic Oscillation (NAO) on the Temperature-Precipitation Relationship in the Iberian Peninsula"

_geosciences, doi:10.3390/geosciences11050211_

Round 1
Reviewer 1 Report
Thank you for the article. I have to admit that to me personal it is not so clear how the analysis can help to shed light on climate reconstructions, so maybe the author can elaborate a bit on the motivation? Since there was no specific funding for this the topic must have "scratched an itch" with the author, maybe he even has a specific problem at hand.
Some comments/questions:
l79 here you use American English (behavior) instead of British English as in the rest of the paper.
l158 the table can be compactified, just listing the years in parenthesis would save a lot of space
l123 Why was the median used here? The NAO/EA indices already have the sign, but you reinterpret them via the median? This way, e.g. in winter you reassign the NAO=-0.2 NAO- phase to NAO+.
l204 Maybe I missed it but I don't know is it (prec_all_seasons - prec_winter) or the other way around? As a reader I can piece this together (my guess is prec_all_seasons - prec_winter) from the plots and the discussion of the NAO+/NAO- properties but I would appreciate an explicit formula. Similarly I think the correlation used in 3.2 should be spelled out explicitely as a fromula somewhere.
l259,368: l259 says that the "EA modulates the influence of the NAO in winter", but this seems only true for precipitation. In Fig 3 the sign follows the NAO phase, not the EA phase. For TX (Fig 5) and to a lesser degree TN (Fig 4) though, EA seems to be the main driver of the observed signs (pos vs neg).
l296: Can you briefly discuss why "cumulative precipitation"? That's what you mean by accumulated, right?
l308 Why switch from triangles pointing up and down to red/black circles? I think triangles are easier to grasp.
l372 I am a bit confused what "switch the subset" here means.
other minor English fixes (disclaimer: I am not a native speaker): l11 patterns -> patterns' l18 of Iberian -> of the Iberian
Reviewer 2 Report
This article presents the impact of NAO and EA on the relationship between temperature and precipitation by using station datasets.
The article is well written and the results are clearly presented. This paper may contribute to the studies in this field. I would recommend accepting this paper with minor changes.
L46 " no role in the average IP temperature" you mean "the variability of the averaged temperature"?
L100 "regimens" should be "regimes"
L152-153 This sentence is not so clear to me
L166 "Table 3 their" should be "Table 3 shows" or "Table 3 presents"
L182-L187 This sentence is too long, maybe it is better to break it down
L289 why is it a first approximation?
Reviewer 3 Report
Review of Exploring Combined Influences of Season EA and NAO ….. Geosciences.
The paper examines the combined roles of the EA and NAO teleconnections on the Iberian Peninsula.
I’m assuming there could be a better way of combining the information depicted in Table 2 in a more concise manner. Perhaps flipping the table such that each teleconnection is on the Y axis with season on the X axis? That would allow the years to be depicted in a more concise manner.
There are only two stations from Portugal. Are those the only two that met the basic data criteria? I’d at least mention this in the text as a significant portion of the western area of the IP is unaccounted for.
I think a PCA would have been another way to analyze the data for this. I’d have liked to have seen more relative to the actual circulation (using a PCA) rather than just using the teleconnection indices, but I can see the justification for what was done.
The paper is well cited and the figures are good.
I think the paper is of good quality, adds to our understanding of the climatology of the region and could be published as depicted. Perhaps a consolidation of the tables, as mentioned above, is in order.
